# Spatio-Temporal Distribution Characteristics of Intangible Cultural Heritage and Tourism Response in the Beijing–Hangzhou Grand Canal Basin in China

**Mo Chen [1], Jiacan Wang [2], Jing Sun [3], Fang Ye [4] and Hongyan Zhang [5,*]**

[1]  School of Humanities, Zhejiang Ocean University, Zhoushan 316000, China; chenmo@zjou.edu.cn
[2]  School of Economics and Management, Zhejiang Ocean University, Zhoushan 316000, China; wjc961216@163.com
[3]  Library, Beijing Institute of Technology, Zhuhai 519088, China; 13330@bitzh.edu.cn
[4]  Development Research Institute, Zhejiang Ocean University, Zhoushan 316000, China; shuang924@126.com
[5]  Library, Zhejiang Ocean University, Zhoushan 316000, China
[*]  Correspondence: zjoulibzhy@zjou.edu.cn

**Abstract:** The Beijing–Hangzhou Grand Canal is renowned for being one of the longest and largest canals in the world. Running from Beijing to Hangzhou (north to south), it connects China's five major water systems and has an important impact on the ecological environment and economy of northern and southern China. It also boasts a large quantity of intangible cultural heritage (ICH). Clarifying the spatio-temporal distribution pattern of ICH in the Beijing–Hangzhou Grand Canal Basin and its influencing factors is essential for the protection and utilization of heritage resources and the formulation of management policies. In this study, 977 national ICH items in the Beijing–Hangzhou Grand Canal Basin are analyzed with the help of ArcGIS spatial analysis technology, SPSS regression analysis, and human geography research methods. The results show that the national ICH in the Beijing–Hangzhou Grand Canal Basin has complete categories but varies in provincial scale, particularly between the north and south parts. According to the analysis using tools such as kernel density estimation, standard deviation ellipse, and the center-of-gravity model, it is clear that the ICH in the Beijing–Hangzhou Grand Canal Basin shows different degrees of sub-type aggregation, varying directional characteristics of each batch of ICH, and a centre of gravity of ICH with a tendency to shift in multiple directions. The main factors affecting the spatio-temporal distribution pattern of ICH in the Beijing–Hangzhou Grand Canal Basin are natural geographical factors, socioeconomic factors, and policy environment factors. Moreover, there is a significant positive correlation between ICH resources and the tourism industry that cannot be ignored. This study provides an important reference for planning the reuse of ICH resource systems in northern and southern China.

**Keywords:** Beijing–Hangzhou Grand Canal; intangible cultural heritage; spatio-temporal distribution; tourism response

## 1. Introduction

Intangible cultural heritage (hereinafter referred to as "ICH") comes in various types, including practices, performances, manifestations, knowledge systems, and skills, as well as related tools, material objects, artware, and cultural sites. All of these are valued by different groups, communities or individuals as cultural heritage [1,2]. In August 2019, the General Office of the State Council released the *Opinions on Further Unleashing the Cultural and Tourism Consumption Potential*, which proposed to promote industrial integration and support the development of ICH-themed tourism and other business formats. The protection and inheritance of ICH and tourism utilization have risen to the national level. As an important part of traditional culture, ICH has long symbolized a carrier of a city's or even a region's spiritual culture. It is the cultural essence passed down

by local people through historical changes, which not only has historical and cultural value but also has social and economic benefits. It is also the core productive force of tourism development [3–7]. In December 2022, General Secretary Xi Jinping issued important instructions on ICH, proposing to take solid steps in the systematic protection of ICH, to promote the transformation and development of China's excellent traditional culture in a creative and innovative manner and to enhance China's global cultural reach.

Foreign ICH protection and research work started early and has achieved fruitful research results in basic fields such as the concept, theoretical connotation, characteristics, value, relationship between ICH and human life, protection and inheritance, and development and utilization [8–13]. Further research mainly focuses on ICH experience and ICH tourism development [14–24]. In recent years, the focus has been extended to diachronic, reflective, and suggestive research on ICH protection, ICH reconstruction and re-innovation, and exploring the relationship between ICH and social and cultural identity in the historical and social dimensions [25–28]. It has made shifts from focusing on the concept of "things" protection to the concept of "people-oriented" protection in a more in-depth way. The research on ICH in China began in 2002, and the research content mainly includes ICH's definition, value, development and utilization, tourism experience, protection and inheritance, revitalization, and ICH and local construction [29–37], among others. The research methods involve tourism, education, history, anthropology, management, psychology, and other fields. Over the past few years, a number of scholars, both at home and abroad, have introduced geographical theories into the study of ICH. Understanding and identifying the spatio-temporal agglomeration characteristics and influencing factors of ICH is the basis for realizing the coordinated and integrated utilization of ICH resources. According to Huang Limin, in essence, ICH is the result of people–environment interaction in a certain region [38]. At present, regional research mostly takes place at the national, provincial, or municipal scales. There is a lack of research on the river basin scale, and there is little quantitative research on ICH resources and the tourism industry. Up to now, the research on ICH in the Beijing–Hangzhou Grand Canal Basin has mainly focused on how to inherit and innovate a certain category of ICH or how to achieve the tourism revitalization of ICH in a certain river section. There is no research exploring the spatio-temporal distribution characteristics, formation mechanism, and tourism response of the ICH in this Basin from a geographical perspective to reveal the regional cultural differences along the Grand Canal [39–41].

The Beijing–Hangzhou Grand Canal is the oldest and longest artificial river in the history of human civilization, with an age of more than 2500 years. It is also the largest in terms of engineering. With its unique functions, it connects China's political center with its economic center. Now, China is building the Grand Canal National Cultural Park, and the protection and inheritance and development and utilization of ICH in the Beijing–Hangzhou Grand Canal Basin is an important part of this endeavor. The concept of a "tourism corridor", which first appeared in the 2016 government report, plays a vital role in many aspects such as natural resource protection and utilization, regional cultural display, regional economic prosperity, and the high-quality and sustainable development of river basins, providing a theoretical basis and methodological guidance for the construction of large-scale linear ICH tourism corridors. The purpose of the tourism corridor is to pursue win-win results in both protection and utilization. Tourism corridors take the special collection of cultural resources as the core attraction, integrate green space, interpretation, products, landscape, transportation, and other elements, and form a comprehensive large-scale tourism destination integrating culture, nature and economy by connecting nodes, routes, and regions [42,43]. In November 2022, General Secretary Xi Jinping proposed that "protecting the cultural heritage along the Grand Canal should be enhanced together with the protection of ecological environment, the protection and restoration of famous cities and towns along the route, the integrated development of cultural tourism, and the transformation and upgrading of canal shipping" during his inspection tour in Jiangsu,

laying a good foundation for the building of Grand Canal Cultural Belt and Grand Canal National Cultural Park.

Based on the nearest neighbor index, kernel density estimation, standard deviation ellipse, and the center-of-gravity model, this paper studies the spatio-temporal distribution characteristics of ICH in the Beijing–Hangzhou Grand Canal Basin and analyzes the main influencing factors that have shaped its unique spatio-temporal pattern. On this basis, regression analysis using SPSS analysis software is used to explore the correlation between the ICH resources and tourism development in the Beijing–Hangzhou Grand Canal Basin. Finally, the study proposes to build an ICH tourism corridor in the Beijing–Hangzhou Grand Canal Basin, to better provide a scientific and feasible basis for promoting regional cultural protection, inheritance, development, and utilization, and to promote the coordinated and sustainable development of culture, economy, and society in the Beijing–Hangzhou Grand Canal Basin.

## 2. Research Data and Methods

### 2.1. Area Coverage

Tracing back to the Spring and Autumn Period, with a total length of about 1794 km, the Beijing–Hangzhou Grand Canal is the "second golden waterway" in China next to the Yangtze River and one of the symbols of Chinese cultural status. The Grand Canal runs from Hangzhou in the south to Beijing in the north, flowing along the way through four provinces and two cities (Zhejiang province, Jiangsu province, Shandong province, Hebei province, Tianjin city, Beijing city) (Figure 1). It also runs through five significant water systems (Haihe River, Yellow River, Huaihe River, Yangtze River, and Qiantang River). Stretching from the Yangtze River Delta to the North China Plain, the Beijing–Hangzhou Grand Canal Basin has been one of the most affluent agricultural areas in China since ancient times, with a flat terrain, intertwined rivers and lakes, and thousands of miles of fertile land. It boasts developed industries and has played a vital part in the political, economic, and cultural exchange between the northern and southern regions of China and in the management of the ecological environment.

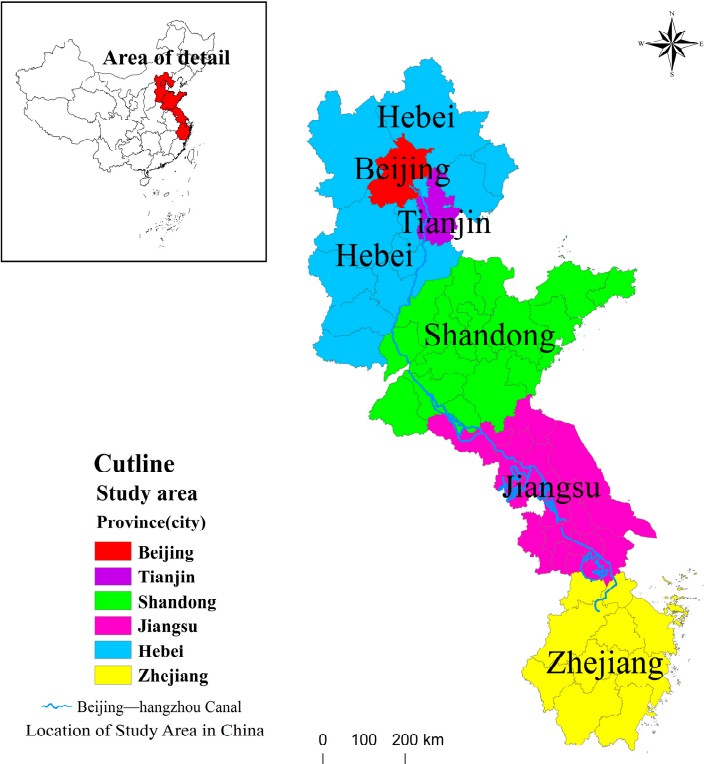

**Figure 1.** Study area.

## 2.2. Data Sources

The ICH-related data in this paper are mainly from the China Intangible Cultural Heritage Network. According to the national list of the ICH of China, they are divided into ten categories. The Ministry of Culture and Tourism of P.R.C. has announced five batches of national intangible cultural heritage lists totaling 1557 items, which are further divided into 3610 sub-items (data obtained by splitting ICH items according to distribution areas). After analyzing the five batches of national-level ICH items on the Beijing–Hangzhou Grand Canal according to the regions in which they are located, 977 ICH items (including extended items) were taken as research samples; the geographic coordinate information of the ICH items was obtained using the coordinate picking system of Baidu Maps; DEM (digital elevation model) data was sourced from geospatial data clouds; Vector data such as administrative divisions, basin boundaries, and river systems were summarized based on standard map services and the data cloud platform of resources and environmental sciences of the Chinese Academy of Sciences; other data concerning population, economy, and policy information were collected from provincial and municipal statistical yearbooks, government work reports, and their official websites.

## 2.3. Research Methods

This paper uses ArcGIS10.8 analysis tools and geographic detectors to explore the spatio-temporal distribution characteristics and influencing factors of the national ICH items in the Beijing–Hangzhou Grand Canal Basin with the help of the following indicators and research methods.

### 2.3.1. Nearest Neighbor Index

The ICH locations are abstractly expressed as spatial point features, and, using the nearest neighbor index, the spatial distribution of these point features is identified. Its calculation formula is [44]:

$$R = \bar{r}/\overline{r_i} \quad \overline{r_i} = \frac{1}{2\sqrt{n/A}} = \frac{1}{2\sqrt{D}} \tag{1}$$

where $\overline{r_i}$ and $r_i$ represent the actual average nearest distance and the theoretical nearest distance, n indicates the quantity of ICH items, and A and D are the area and the point density, respectively. When R > 1, it means that ICH items tend to be evenly distributed; when R = 1, it indicates a random distribution of ICH items; when R < 1, it is a sign of aggregated distribution.

### 2.3.2. Kernel Density Analysis

The Kernel Density Estimation method is used to measure the degree of agglomeration and the main gathering places of ICH items in various provinces and cities in the Beijing–Hangzhou Grand Canal Basin, and its calculation formula is [44]:

$$P_n(X_i) = \frac{1}{n * h_n} \sum_{j=1}^{n} K\left\{ \frac{x_i - x_j}{h_n} \right\} \tag{2}$$

where $k\left( \frac{x_i - x_j}{h_n} \right)$ is the kernel function, n is the total number of ICH samples, h stands for bandwidth ($h > 0$), and ($x_i - x_j$) is the distance between the valuation point and the ICH sample point.

### 2.3.3. Standard Deviation Ellipse

Standard deviation ellipse analysis takes the standard deviation of X and Y axes and the mean center as the basic parameters and is a method of quantifying the spatial distribution range of geographical elements and analyzing their patterns and direction of transformation. The analysis results can intuitively express the spatial distribution pattern

and general direction of different batches of ICH items in the Beijing–Hangzhou Grand Canal Basin, and its calculation formula is [45]:

$$X = \frac{\sum_{i=1}^{n} x_i}{n} \quad Y = \frac{\sum_{i=1}^{n} y_i}{n} \tag{3}$$

where $X_i$ and $Y_i$ are the coordinates of ICH sample points, while n indicates the total number of ICH points of a certain type.

### 2.3.4. Center-of-Gravity Model

The center-of-gravity model is an important analytical tool to explore the spatial evolution of regional geographic features; it can clearly reflect the spatial differences and dynamic processes of regional geographical phenomena by calculating the deviation distance and direction of the center-of-gravity of geographical features. Its calculation formula is [46]:

$$\overline{x} = \sum_{i=1}^{n} M_i X_i / \sum_{i=1}^{n} M_i \quad \overline{y} = \sum_{i=1}^{n} M_i Y_i / \sum_{i=1}^{n} M_i \tag{4}$$

where $(\overline{x}, \overline{y})$ is the coordinate of the center of gravity of the regional geographic feature; $(\overline{X}, \overline{Y})$ refers to the geographic coordinate of the *i*th sub-region; $M_i$ indicates the attribute value of a geographic feature in the *i*th sub-region.

## 3. Results and Analysis

### 3.1. Quantitative Differences in the Distribution of ICH Items in the Beijing-Hangzhou Grand Canal Basin

From the distribution of ICH items in each province and municipality along the Beijing–Hangzhou Grand Canal (Table 1), Zhejiang has the largest number of ICH items, with up to 257 (accounting for 26.31%); Shandong ranks second, with 186 items (accounting for 19.04%); Beijing, Hebei, and Jiangsu are very close, with 164, 162, and 161 ICH items, respectively, accounting for 16.79%, 16.58%, and 16.48%; and Tianjin has only 47 ICH items (accounting for 4.81%), which is the lowest. In terms of ICH categories, the largest number of ICH items is in the category of traditional craftsmanship, up to 193 items (19.75%); the categories with more than 100 ICH items are traditional fine arts (129, 13.2%) and traditional drama (127, 13%); the categories with 75 to 100 ICH items are folk customs, traditional music, folk literature, folk vocal art forms, and traditional sports, competitive sports, and acrobatics, with 97, 87, 77, 75, and 75 items, respectively. The number of ICH items related to traditional dance and traditional medicine is relatively small, only 61 (6.24%) and 56 (5.73%), respectively.

**Table 1.** The number of ICH items in each province or municipality by category: geographical distribution.

| Province (Municipality/ Autonomous Region) | Folk Literature | Traditional Music | Traditional Dance | Traditional Drama | Folk Vocal Art Forms | Traditional Sports, Competitive Sports, and Acrobatics | Traditional Arts | Traditional Craftsmanship | Traditional Medicine | Folk Customs | Total/ Proportion (%) |
|---|---|---|---|---|---|---|---|---|---|---|---|
| Beijing | 9 | 5 | 9 | 7 | 8 | 14 | 22 | 53 | 20 | 17 | 164/16.79% |
| Tianjin | 1 | 5 | 1 | 4 | 7 | 8 | 3 | 8 | 8 | 2 | 47/4.81% |
| Hebei | 5 | 23 | 11 | 36 | 9 | 24 | 15 | 21 | 4 | 14 | 162/16.58% |
| Shandong | 27 | 18 | 13 | 33 | 13 | 15 | 28 | 19 | 6 | 14 | 186/19.04% |
| Zhejiang | 24 | 15 | 18 | 25 | 28 | 12 | 30 | 54 | 12 | 39 | 257/26.31% |
| Jiangsu | 11 | 21 | 9 | 22 | 10 | 2 | 31 | 38 | 6 | 11 | 161/16.48% |
| Total | 77 | 87 | 61 | 127 | 75 | 75 | 129 | 193 | 56 | 97 | 977/100% |

*3.2. Types of Spatial Distribution of ICH Items in the Beijing-Hangzhou Grand Canal Basin*

The overall number of ICH items in the Beijing–Hangzhou Grand Canal Basin was processed using the Average Nearest Neighbor tool in the ArcGIS 10.8 software; the average nearest neighbor index obtained was R = 1.621, with R > 1 and P = 0 (Table 2), which determined that the spatial layout type of the overall ICH items in the Beijing–Hangzhou Grand Canal Basin was evenly distributed. From the perspective of each category, the nearest neighbor index of the ten categories of ICH items was less than 1, which determined that the spatial distribution pattern of ICH by category was clustered distribution. Among them, the nearest neighbor index of ICH items in the categories of traditional craftsmanship, traditional sports, competitive sports and acrobatics, folk customs, and folk vocal art forms was between 0.4–0.6, showing significant spatial agglomeration, with the agglomeration characteristics of ICH items in the traditional craftsmanship category being the most obvious; in contrast, the nearest neighbor index of ICH items in the categories of traditional fine arts, traditional dance, traditional drama, traditional medicine, traditional music and folk literature was between 0.6–0.9, and the agglomeration was weak.

**Table 2.** Nearest neighbor index of ICH items as a whole and by category.

| Type | Total | Traditional Craftsmanship | Traditional Arts | Traditional Sports, Competitive Sports, and Acrobatics | Traditional Dance | Traditional Drama | Traditional Medicine | Traditional Music | Folk Literature | Folk Customs | Folk Vocal Art Forms |
|---|---|---|---|---|---|---|---|---|---|---|---|
| R | 1.621 | 0.403 | 0.612 | 0.563 | 0.868 | 0.615 | 0.629 | 0.621 | 0.642 | 0.549 | 0.504 |
| Z | 8.66 | −15.86 | −8.43 | −7.24 | −1.97 | −8.31 | −5.29 | −6.75 | −6.02 | −8.50 | −8.22 |
| P | 0 | 0 | 0 | 0 | 0.05 | 0 | 0 | 0 | 0 | 0 | 0 |

*3.3. Spatial Distribution Characteristics of ICH in the Beijing-Hangzhou Grand Canal Basin*

3.3.1. Distribution Characteristics of ICH Items as a Whole

Based on ICH geographic coordinates, kernel density analysis using ArcGIS 10.8 spatial analysis tool was conducted, and the five-level natural breaks classification method was used to establish the overall ICH density analysis map of the Beijing–Hangzhou Grand Canal Basin (Figure 2). (1) The spatial distribution of ICH in the Beijing–Hangzhou Grand Canal Basin has a significant agglomeration trend, showing a pattern of "gathering in the north and south, scattering in the middle" and an obvious structure of "three cores, two belts and three verticals". (2) "Three cores" include one high-density core area and two sub-density ones. To be specific, Beijing and the northern area of Langfang are the core centers that form the high-density core area, and the northeast of Baoding and the northwest of Tianjin are radiated to form the sub-density core area with the widest coverage. Moreover, the second sub-density core area is formed with the four northern cities of Zhejiang as the center; "Two belts" include two low-density distribution belts covering Beijing, Baoding, Langfang, Tianjin, Cangzhou, and urban agglomerations in southern Jiangsu and northern Zhejiang, with medium core density values. "Three verticals" refer to the areas with low kernel density value and a scattered distribution of ICH, including the Beijing-Tianjin-Hebei belt, the Shandong belt, and the Jiangsu-Zhejiang belt. (3) The overall core gathering area of ICH in the Beijing–Hangzhou Grand Canal Basin is mainly located in the political center and economic center, which reflects that both the political and cultural environment and the urban economic development level have a significant impact on ICH spatial distribution.

3.3.2. Distribution Characteristics of ICH by Category

The kernel density processing maps of ICH items in the Beijing–Hangzhou Grand Canal Basin were established by category, and it was found that in this Basin, different categories of ICH have different agglomeration areas and that they show great differences (Figure 3). The ICH items included in the categories of folklore, traditional craftsmanship,

and folk vocal art forms are gathered in the Beijing-Hebei region and Zhejiang province; the ICH items of traditional sports, competitive sports and acrobatics, and traditional medicine are gathered in the Beijing-Hebei region; traditional music, traditional fine arts and traditional dance are gathered in the Beijing-Hebei region and Jiangsu and Zhejiang regions; folk literature and traditional drama are clustered in multiple regions.

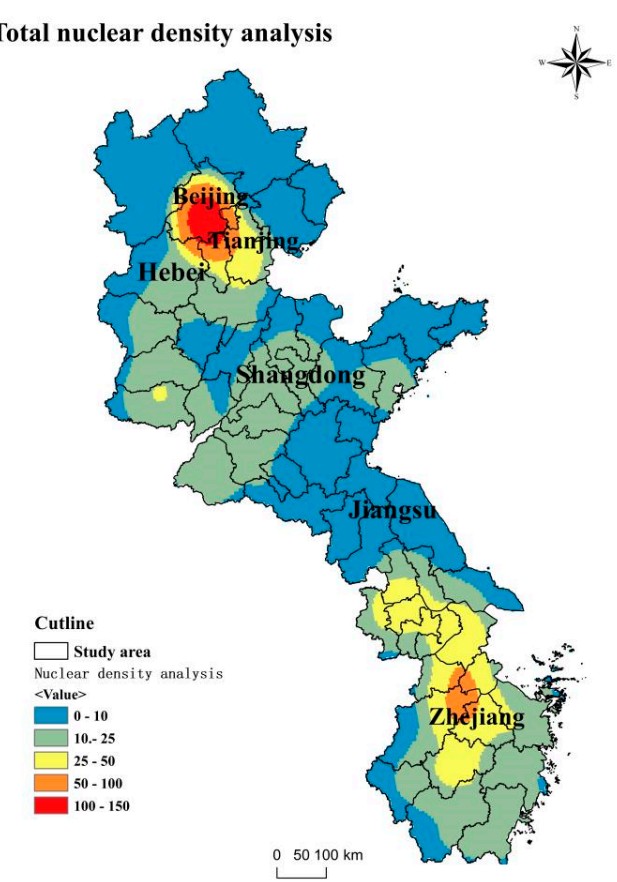

**Figure 2.** Kernel density processing map of ICH in the Beijing–Hangzhou Grand Canal.

*3.4. Temporal Characteristics*

As of 2021, the State Council has published five batches of national representative lists of ICH (Table 3). In order to analyze the trend and direction characteristics of the distribution of different batches of ICH in the Beijing–Hangzhou Grand Canal Basin, this paper uses the kernel density analysis, standard deviation ellipse analysis and mean center analysis tools in the spatial analysis toolbox of ArcGIS 10.8 software to dynamically reveal the dispersion degree of the first to fifth batches of ICH in the spatial distribution and further dynamically reveal the temporal evolution law in the distribution direction of each batch of ICH in the Beijing–Hangzhou Grand Canal Basin. Figure 4 shows that there are differences in the temporal and spatial evolution characteristics of the five batches of ICH items and that the first batch of ICH items are mainly concentrated in Beijing, Tianjin, Langfang, Cangzhou, and Baoding, as well as Suzhou, Hangzhou, Huzhou, Jiaxing, and Shaoxing, and the ICH items in the rest of the areas are scattered. The second batch of ICH items has obvious agglomeration characteristics and is highly concentrated in the Beijing-Tianjin-Hebei region; the distribution of the third batch of ICH items is similar to the first batch, but the density in northern Zhejiang has increased significantly, and the geographical scope is larger. The overall diffusion trend of the fourth batch of ICH items is obvious, and the ICH covers the widest area; the intensity of the fifth batch of ICH items has increased rapidly, forming the only polar core area with Beijing as the core and radiating to Tianjin, Langfang, and Baoding, and the distribution range has decreased. In general, the five batches of ICH items showed a trend of "starting high and going low" [47].

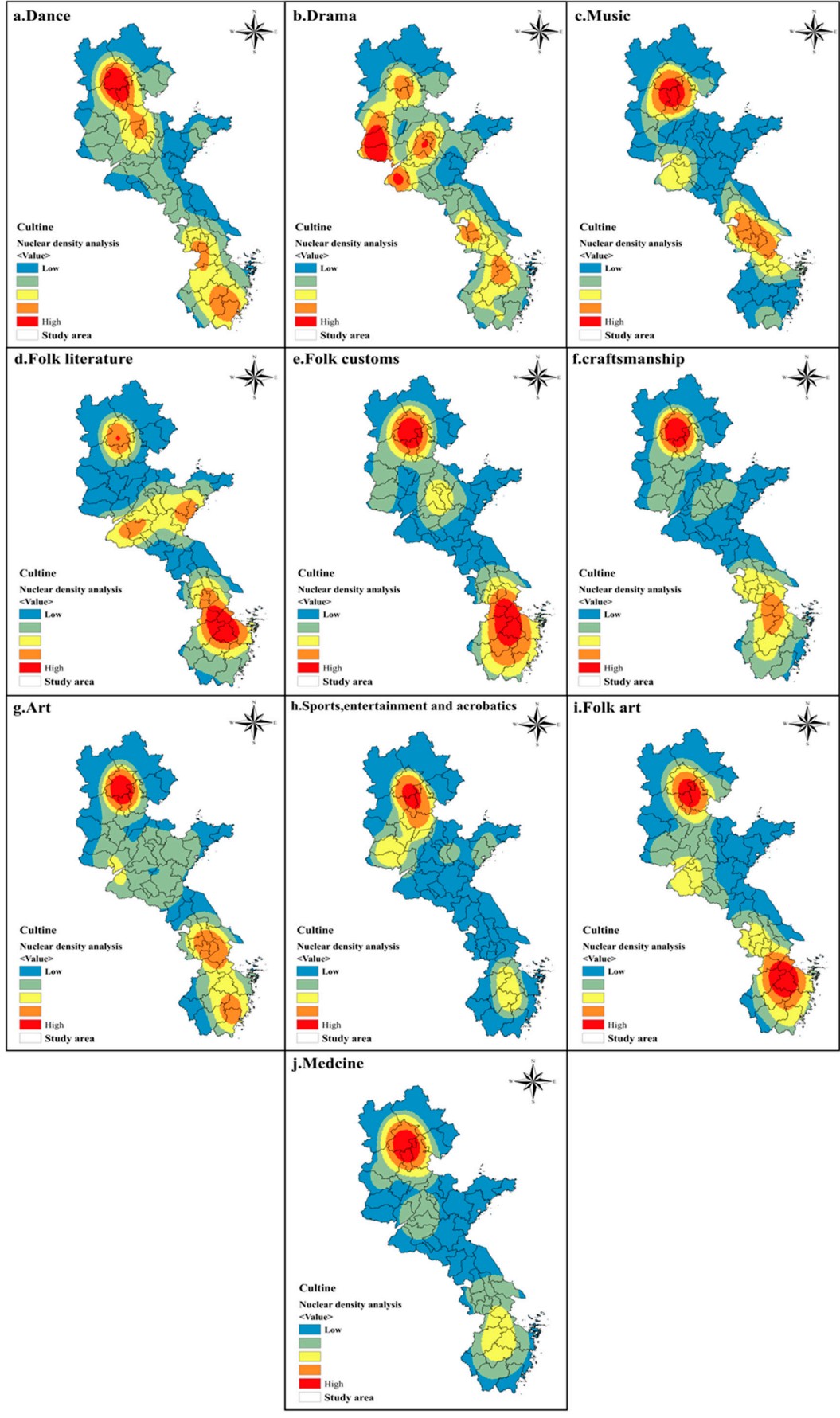

**Figure 3.** Kernel density analysis of ICH by category.

**Table 3.** The number of each batch of national-level ICH items in various provinces and municipalities.

| Province | First Batch | Second Batch | Third Batch | Forth Batch | Fifth Batch | Total |
|---|---|---|---|---|---|---|
| Beijing | 32 | 71 | 18 | 18 | 25 | 164 |
| Tianjin | 7 | 10 | 5 | 11 | 14 | 47 |
| Hebei | 39 | 78 | 15 | 16 | 14 | 162 |
| Shandong | 27 | 93 | 33 | 20 | 13 | 186 |
| Jiangsu | 37 | 62 | 27 | 19 | 16 | 161 |
| Zhejiang | 46 | 97 | 60 | 30 | 24 | 257 |
| Total | 188 | 411 | 158 | 114 | 106 | 977 |

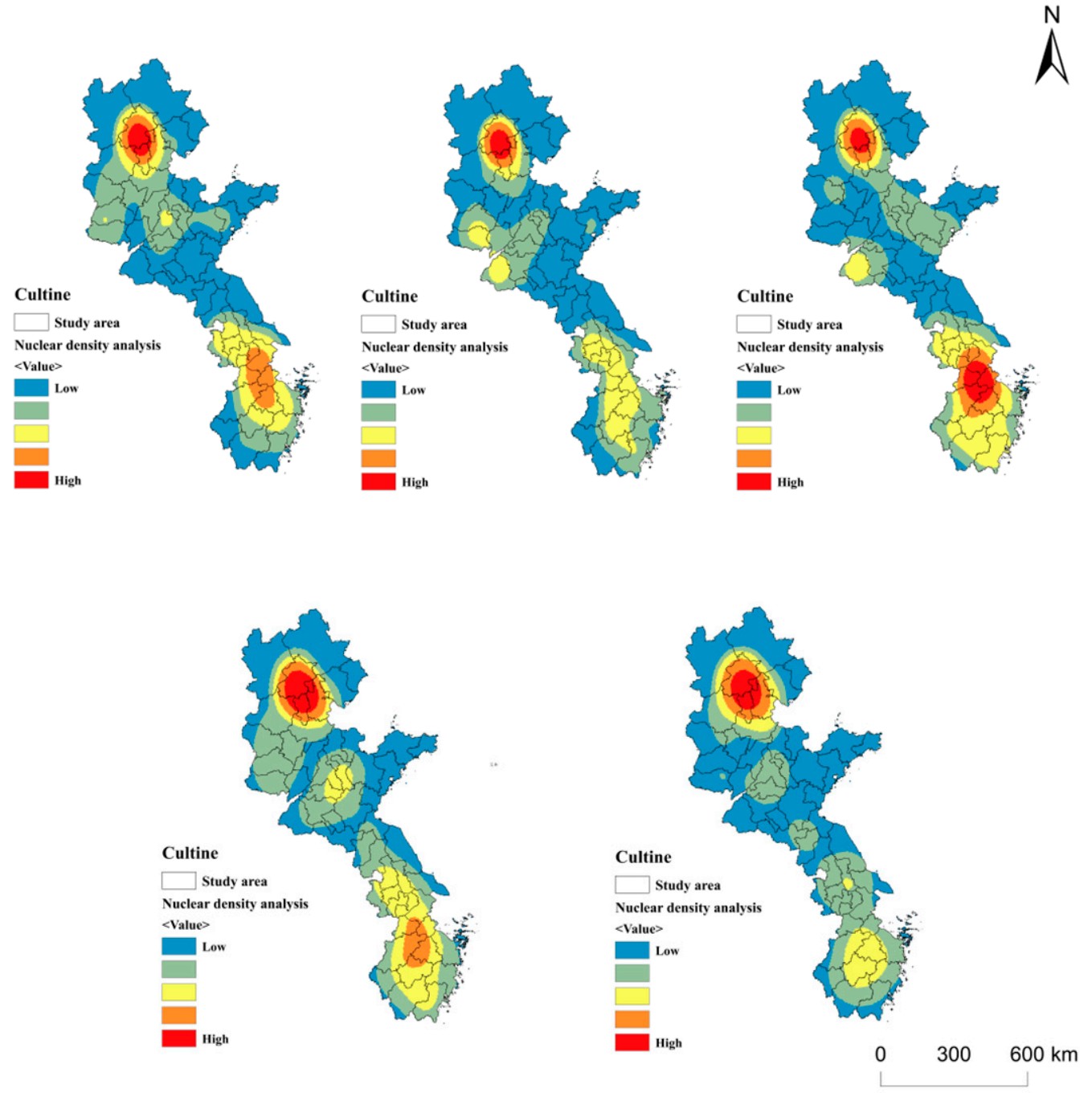

**Figure 4.** Kernel density map of each batch of ICH items.

Through the analysis of the standard deviation ellipse and the center-of-gravity model (Figure 5), it is found that the overall ICH is distributed in the direction of "northwest-southeast", and the center of gravity falls in Linyi City, Shandong Province. The direction characteristics of the five batches of ICH items are consistent with the overall direction, all showing a "northwest-southeast" distribution, but there are great differences in the center of gravity of each batch of ICH items, showing a trend of "northwest-southeast-northwest-northwest". The center of gravity of the first batch of ICH items is located in Linyi city; the center of gravity of the second batch has shifted slightly to the northwest but still falls in Linyi city; the center of gravity of the third batch has shifted significantly to the southeast and falls at the junction of Huai'an city and Suqian city; the center of gravity of the fourth batch has moved northwest and returned to Linyi city; the center of gravity of the fifth batch continues to move northwest and falls at Jinan city. The main axis of the standard deviation first becomes larger and then becomes smaller, and the ICH items are polarized in the main distribution direction, with the spatial distribution direction becoming more and more obvious. Overall it goes through a process of concentration and then dispersion. This is because the Beijing-Hebei region has always been a highly concentrated area of ICH resources.

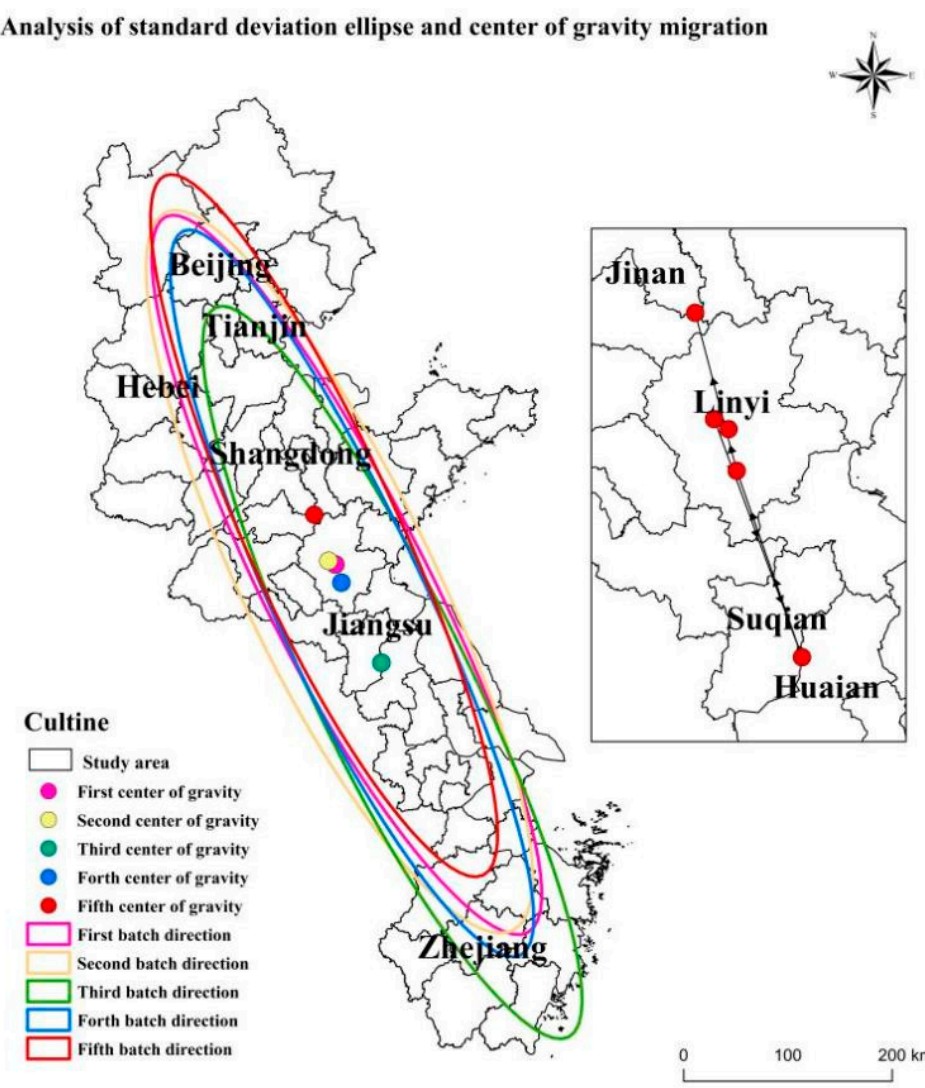

**Figure 5.** Standard deviation ellipse and center-of-gravity analysis of ICH items by batch.

### 3.5. Reasons for the Formation of Spatio-Temporal Distribution Characteristics of ICH

As a historical "living" culture, ICH is a cultural phenomenon created by the people in a specific historical period and region, and it is deeply rooted in the natural geographical environment and profound cultural environment in which it grows [48]. According to existing studies, the specific factors affecting the spatio-temporal distribution characteristics of ICH often include topography and landform, river system, population, GDP, urbanization rate, number of cultural places, ethnicity, number of policies, etc.

#### 3.5.1. Natural Geographic Factors

Topographical features and river systems shape the environment and regional cultural characteristics, having a certain impact on the spatio-temporal distribution differences of ICH items. Since ancient times, ancestors have followed the principle of "living by the water". There is a high degree of matching between river network density and ICH distribution in the Beijing–Hangzhou Grand Canal Basin (Figure 6), but it also shows a low degree of differentiation. This is because the cities along the route are close to water systems. In addition to the hills and mountains in northern Hebei, the landform in four provinces and two municipalities in the Beijing–Hangzhou Grand Canal Basin is mainly plains that facilitate population migration and cultural exchange, thereby promoting ICH formation and development. However, there are a small number of ICH items in Hebei, as the topography of different cities in the region features low variation. In summary, natural geography has a certain influence on the formation of the spatial and temporal distribution characteristics of ICH in the Beijing–Hangzhou Grand Canal Basin, but the influence is not obvious.

#### 3.5.2. Socio-Economic and Policy Environmental Factors

First of all, the socio-economic behaviors of humans have a crucial impact on ICH spatial and temporal distribution. People in areas with higher levels of economic development usually begin to pursue spiritual and cultural consumption needs and will actively participate in the learning, experience, and inheritance of ICH. Beijing-Tianjin and northern Jiangsu and northern Zhejiang are highly developed areas in terms of GDP in the Beijing–Hangzhou Grand Canal Basin, which are also high-density ICH gathering areas, while the central part of the Basin has an average level of urban economic development, so it mainly has low-density ICH gathering areas. Secondly, cultural factors and demographic factors also have a great influence on the spatial and temporal distribution of ICH. For example, Beijing, Hangzhou, and Nanjing are the ancient capitals of various dynasties, with a long history and profound cultural heritage, and they have formed a unique regional culture and a rich variety of ICH items under the collision and integration of the northern and southern cultures in various periods. At the same time, regions with more museums, cultural centers, and traditional villages are often more suitable for the survival and development of ICH, reflecting the importance attached to cultural development in the region and emphasizing the protection and inheritance of culture. Places with large populations usually experience cultural collisions and exchange, and the development of ICH is more active. In addition, urbanization rate, transportation accessibility, and industrialization level may also influence the spatio-temporal distribution of ICH. These indicators reflect the positive effect of the optimization and improvement of a region's industrial structure and infrastructure system on the exchange and spread of ICH, which is conducive to the protection of ICH. Last but not least, the impact of policy environment factors on the spatial and temporal distribution of NRM cannot be ignored. Although the degree of protection and development of ICH varies from region to region, all provinces and cities in the Basin place a high priority on supporting ICH-related work. Governments at all levels adhere to the protection principle of "government leadership and social participation" and invest more funds in the construction of ICH on the basis of improving the legal protection system of ICH.

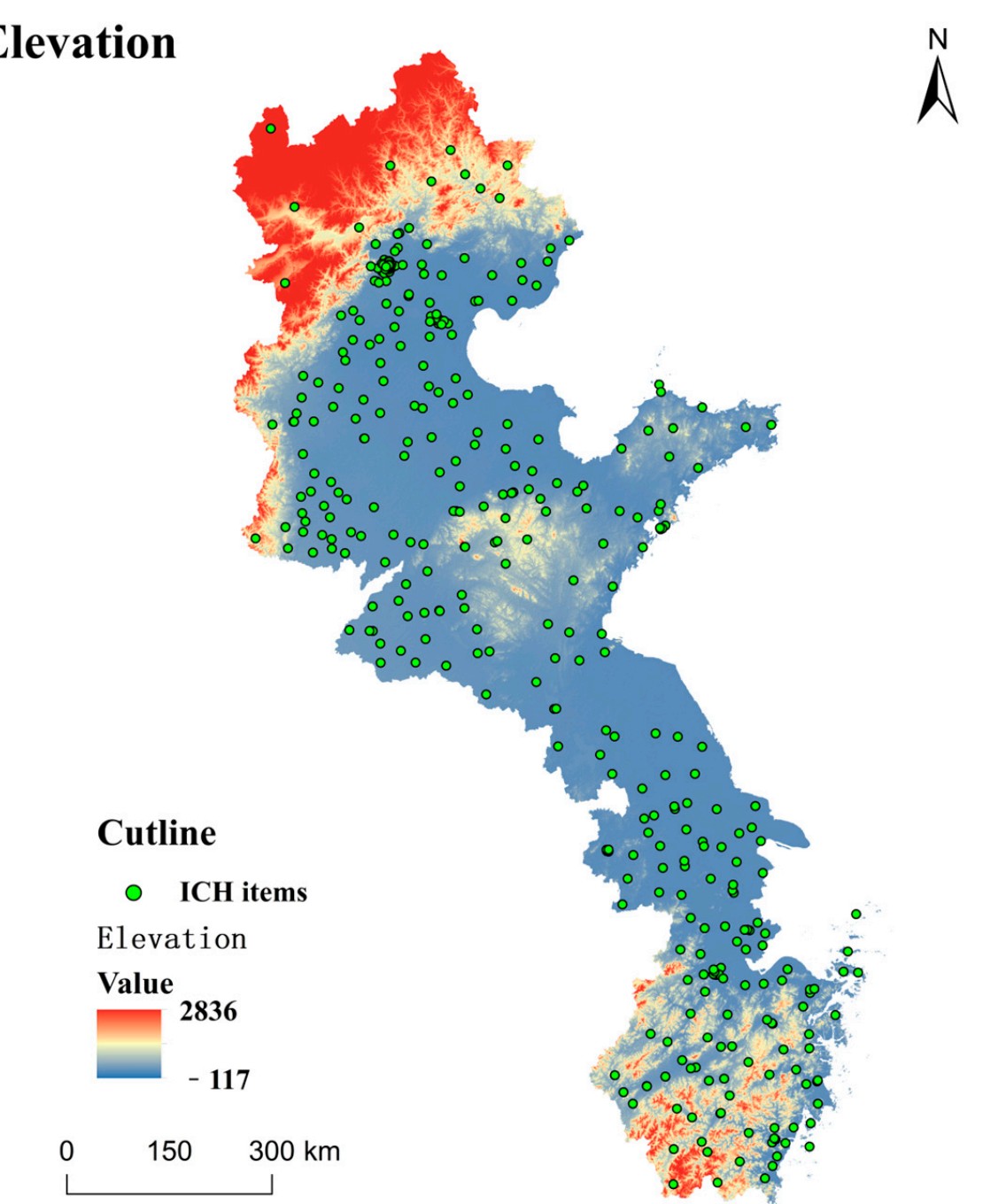

**Figure 6.** Elevation, slope direction, and slope gradient of Beijing–Hangzhou Grand Canal.

*3.6. Correlation between the Spatio-Temporal Distribution of ICH and Tourism Development*

The above expounds the temporal evolution characteristics, spatial structure characteristics, and causes of ICH in the Beijing–Hangzhou Grand Canal Basin and sorts out the endowment of ICH resources in the Basin in detail. In order to further explore the relationship between the protection, development, and inheritance of ICH and tourism development, the gross tourism revenue, gross domestic tourism revenue, domestic tourist arrivals, inbound tourist arrivals, foreign currency earnings from tourism, and quantity of A-grade scenic spots (Table 4) in the four provinces and two municipalities in the Basin in 2021 were selected as independent variables, and the number of ICH items as dependent variables, and the quantitative analysis was carried out through the SPSS software [49,50].

**Table 4.** Related data of ICH and the tourism industry by province and municipality in 2021.

| Province/ Municipality | Number of ICH Items | Gross Tourism Income | Gross Domestic Tourism Revenue | Domestic Tourist Arrivals | Inbound Tourist Arrivals | Foreign Exchange Income from Tourism | Number of A-Grade Scenic Spots |
|---|---|---|---|---|---|---|---|
| Beijing | 164 | 4166.2 | 4138.5 | 25,448.3 | 24.5 | 4.3 | 243 |
| Tianjin | 47 | 1352.9 | 1331.42 | 14,100 | 17.13 | 3.34 | 100 |
| Hebei | 162 | 4424.42 | 3674.67 | 37,952.5 | 7.9 | 0.3 | 490 |
| Shandong | 186 | 8278.6 | 8278.6 | 73,052.2 | 52.8 | 0 | 1193 |
| Jiangsu | 161 | 11,672.72 | 11,593.9 | 70,617.6 | 61.9 | 11.43 | 615 |
| Zhejiang | 257 | 10,687 | 6171 | 39,943 | 42.84 | 2.04 | 798 |

Note: Due to the COVID-19 epidemic, Shandong Province suspended the accounting of inbound tourism data in 2021.

The results (Table 5) show that the significance level between each element of the tourism industry and the ICH resources is sig = 0 < 0.05, and the coefficient test between the two is significant, indicating that ICH can effectively predict the variation of other dependent variables. In addition, the correlation coefficients between ICH resources and gross tourism revenue, gross domestic tourism revenue, domestic tourist arrivals, inbound tourist arrivals, foreign currency earnings from tourism, and the quantity of A-grade scenic spots in the Beijing–Hangzhou Grand Canal Basin are all 1, indicating that the regression equation has the best degree of interpretation of the dependent variable, showing a perfect fitting effect, that is, there is an extremely significant positive correlation between the ICH resources of the Beijing–Hangzhou Grand Canal Basin and tourism development.

**Table 5.** Regression analysis of ICH resources and the tourism industry.

|  | Gross Tourism Income | Gross Domestic Tourism Revenue | Domestic Tourist Arrivals | Inbound Tourist Arrivals | Foreign Exchange Income from Tourism | Number of A-Grade Scenic Spots |
|---|---|---|---|---|---|---|
| sig | 0 | 0 | 0 | 0 | 0 | 0 |
| $R^2$ | 1 | 1 | 1 | 1 | 1 | 1 |

$R^2$ In the context of China's promotion of cultural and tourism integration, as a revitalized cultural tourism resource with an appreciation, experience, and entertainment nature, ICH can provide a heritage tourism resource guarantee for the development of tourism along the Beijing–Hangzhou Grand Canal, shape a tourism brand image with historical and practical significance, and build a new format, new model and new core for the development of the tourism industry in the areas along the Beijing–Hangzhou Grand Canal. At the same time, most of the provinces (cities) along the Beijing–Hangzhou Grand Canal have a strong tourism industry, so in order to make further breakthroughs in the tourism market and create strong provinces (cities) in cultural tourism, they need to be upgraded with the integration of ICH resources into local scenic spots.

*3.7. Construction of the Beijing-Hangzhou Grand Canal ICH Tourism Corridor*

As an important cultural tourism resource that can break through administrative boundaries and is not limited by geography, ICH plays an important role in the construction of the Yangtze River, Yellow River, and Grand Canal National Cultural Parks. The construction of linear cultural heritage belt focuses on the integration of regional cultural resources, maximizing the efficiency of the cross-regional coordinated development of resources, and jointly building a shared cultural community. In 2021, the General Office of the CPC Central Committee issued *Opinions on Further Strengthening the Protection of Intangible Cultural Heritage*, which pointed out that it is necessary to promote the integrated and high-quality development of ICH and tourism and create a multi-dimensional intangible cultural heritage tourism model.

Firstly, on the basis of understanding the spatio-temporal distribution characteristics and formation mechanism of ICH in the Basin, the agglomeration effect of ICH should be analyzed to enhance the understanding, cognition, and recognition of ICH among the general public and further promote local governments' awareness of macro issues such

as "what are ICH resources", "what NRM resources include", and "why NRM resources are formed", so as to lay a foundation for the promulgation and implementation of ICH-supporting policies. Secondly, it is required to give full play to the unique advantages of the four provinces and two municipalities in terms of natural geography, cultural ecology, economy, and policies, as well as a good brand image for the tourism market, and effectively push forward the high-quality development of integrating ICH resources with the tourism industry on the basis of protection. It is advised to take Beijing-Tianjin and Southern Jiangsu-Northern Zhejiang as the core, to construct a bipolar linear cultural heritage tourism belt and form "agglomeration points", "agglomeration lines", and "agglomeration surfaces" based on the distribution characteristics, so as to complete the efficient integration of resources spatially. Finally, drawing on the advanced management mode and experience of international regional resource integration, a Beijing–Hangzhou Grand Canal Intangible Cultural Heritage Tourism Corridor Alliance can be established to build a linear cultural belt of ICH tourism, clustered intangible cultural heritage tourism cultural complexes, and intangible cultural heritage tourism cultural scenic spots at the macro, meso, and micro levels. The protection and tourism development should go hand in hand to create a linear cultural tourism destination with international exemplary significance [42,51,52].

## 4. Discussion and Conclusions

### 4.1. Discussion

The Beijing–Hangzhou Grand Canal stretches from China's economic center in the south to the political center in the north, with a superior location and a unique humanistic ecology and political and economic environment. The spatio-temporal distribution pattern and tourism correlation of ICH in the Beijing–Hangzhou Grand Canal Basin are typical and representative. Through analysis, it can be found that the ICH at the north and south ends of the Beijing–Hangzhou Grand Canal shows a strong agglomeration, while the middle area also shows a certain spatial agglomeration in specific ICH categories. The density of ICH distribution in areas close to the Beijing–Hangzhou Grand Canal is greater than that of other areas, indicating that there is a direct correlation between the formation and development of ICH in the Basin and the distance from the Canal. Therefore, in the future management of the canal basin area, the ICH should be fully considered.

The study of the formation and development of ICH further illustrates the influence between factors such as natural geography, cultural ecology, and socio-economic and political environment. The rich ICH resources also reflect the importance of the Beijing–Hangzhou Grand Canal in historical, economic, and social development. In the further development of ICH in the future, attention should be paid not only to its historical, cultural, and economic value, but also to its reuse value and its relationship with the external environment [41]. The ecosystem of ICH is a unit of the entire cultural ecosystem, which includes not only the intrinsic resource endowment, geographical spatial distribution, specific environmental space, and accumulated culture, brand, and image of ICH, but also the complex relationship generated by interaction with external units, including external relationship factors such as the demand status of ICH, audience groups, technical conditions, and relevant policies and systems. Therefore, the ecological niche of ICH is mainly composed of two aspects: the ontological conditions of ICH and its external relations. According to the ecological niche theory, the ecological niche width of ICH resources can be expanded, the ability of ICH in the Basin to adapt to the surrounding environment can be improved, the external development conditions of ICH can be gradually improved, and the needs of ICH audiences and inheritors can be accurately grasped, so as to win the ecological niche advantage of resource demand and, finally, enrich the level of spatio-temporal ecological niches and reasonably expand the audience and dissemination scope of ICH [53–55].

The above analysis provides policy enlightenment for the protection, inheritance and reuse of ICH in the Beijing–Hangzhou Grand Canal Basin. It is suggested to establish a national center for the protection of ICH in the Beijing–Hangzhou Grand Canal Basin,

adopt a phased and differentiated development strategy, and give play to the joint role of "point-line-area". Secondly, we should attach importance to the regeneration, nurturing, and sustainable development of ICH in the areas along the Beijing–Hangzhou Grand Canal, introduce and implement relevant protection policies, build a digital management service platform for ICH, and improve the collection, storage, management, protection, display, and utilization of ICH in regions along the route. Finally, it is necessary to promote the efficient integration of ICH resources and the tourism industry in the Basin according to local conditions and encourage the reuse of ICH resources through a mature tourism market. At the same time, with cultural and creative design as the core, efforts should be made to deeply explore the value of ICH, create an ICH cultural and creative tourism product system that can not only meet the needs of public life but also conform to contemporary aesthetics, and promote the revitalization of ICH.

### 4.2. Conclusions

Based on the ArcGIS software, the nearest neighbor index, kernel density analysis, standard deviation ellipse, the center-of-gravity model and SPSS regression analysis, this paper studies the spatio-temporal distribution characteristics of ICH in the Beijing–Hangzhou Grand Canal Basin and its relationship with tourism response. The main conclusions are as follows:

1. The overall distribution of ICH is concentrated mainly at the north and south ends of the Beijing–Hangzhou Grand Canal but scattered in the middle areas. This distribution pattern is mainly closely related to the economic development and political status of the region. The distribution characteristics of each batch of ICH items are different. On the whole, it has undergone an evolution process of "first concentration, then dispersion and then stability", and the focus of the five batches of ICH items shows a trend of "northwest-southeast-southwest-northwest".

2. The ten categories of ICH items are closely linked to the development history of the regions where they are located and have formed different agglomeration characteristics in different regions. The main agglomeration area is the Beijing-Tianjin area, while the northern Zhejiang and southern Jiangsu areas form a sub-density concentration area on specific types of ICH resources.

3. The spatio-temporal distribution characteristics of ICH along the Beijing–Hangzhou Grand Canal were formed under the comprehensive influence of various factors, and the main influencing factors are natural geographical factors, socio-economic factors, and policy and environmental factors.

4. There is a significant positive correlation between ICH and tourism development in the Beijing–Hangzhou Grand Canal Basin. In the future development of cultural tourism, regions along the Beijing–Hangzhou Grand Canal should attach great importance to promoting the in-depth integration and development of ICH resources and the tourism industry, dig deep into the connotation of ICH, strengthen the construction of ICH tourism brand image, concentrate efforts to develop ICH cultural and creative industries, increase multi-channel publicity for ICH tourism, and make every effort to build an ICH tourism corridor alliance with international popularity and exemplary significance. ICH can be reused in the context of cultural and tourism integration, and can be better protected and inherited.

In summary, this study provides a comprehensive analysis of the ICH in the Beijing–Hangzhou Grand Canal Basin and its relationship with tourism, which provides a useful reference for local governments in the protection of ICH and the integration of culture and tourism.

**Author Contributions:** Conceptualization, M.C.; data collection, J.W. and J.S.; methodology, J.W.; data analysis, M.C. and H.Z.; software, H.Z. and J.W.; writing—original draft preparation, M.C.; resources, J.S. and F.Y.; review and editing, H.Z. and M.C.; supervision, H.Z.; funding acquisition, H.Z. and F.Y. All authors have read and agreed to the published version of the manuscript.

**Funding:** The research was founded by National Social Science Foundation of China (grant number 21&ZD233); Youth Key Project of Major Research Projects in Humanities and Social Sciences in Universities, Zhejiang, China (grant number 2023QN086).

**Institutional Review Board Statement:** Not applicable.

**Informed Consent Statement:** Not applicable.

**Data Availability Statement:** Not applicable.

**Acknowledgments:** We express our gratitude to the anonymous reviewers and editors for their professional comments and suggestions.

**Conflicts of Interest:** The authors declare no conflict of interest.

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
