# Peer review of "Spatio-Temporal Distribution Characteristics of Intangible Cultural Heritage and Tourism Response in the Beijing–Hangzhou Grand Canal Basin in China"

_sustainability, doi:10.3390/su151310348_

Round 1
Reviewer 1 Report
Dear Authors,
Congratulations for your investigation that put-on evidence the need to valorize the intangible heritage as tourism resource, but especially prevent negative impacts with protection measures.
In general, it is a very interesting study indeed, but a few comments are appropriate:
- in introduction section you can explore some considerations on the literature about the relevance of ICH as touristic resource;
- in the methodology section include the list of ICH thar are studied (sub-section 2.2);
- If it is possible, introduce some photos to characterize the case study area in sub-section 2.1.
- in figures 2 to 5, highligthed the Grand Canal Basin localization to localize the reader;
Overall you did a good job!
Best wishes
Author Response
Dear Reviewer,
Thank you for your valuable feedback on my manuscript. I have carefully read your suggestions and made corresponding revisions to the manuscript (Please see the attachment). Here are my responses to your suggestions:
Point 1:- in introduction section you can explore some considerations on the literature about the relevance of ICH as touristic resource;
Response 1:Thank you very much for your valuable feedback on our paper. We sincerely appreciate your suggestions. We have taken into consideration several literature references related to intangible cultural heritage tourism in our work, and we plan to further explore this topic in our future research.
Point 2:- in the methodology section include the list of ICH thar are studied (sub-section 2.2);
Response 2:Thank you for your valuable feedback on my paper. I have carefully considered your suggestion regarding the inclusion of a detailed list of the intangible cultural heritage projects in the Jing-Hang Great Basin. However, due to the large number of projects involved, it would be challenging to accommodate an exhaustive list within the limited space available in the abstract.
Point 3:If it is possible, introduce some photos to characterize the case study area in sub-section 2.1.
Response 3:Thank you for your valuable suggestions. We appreciate your feedback and have taken it into careful consideration. We will make the necessary revisions to present a clearer diagram of the research area.
Point 4:- in figures 2 to 5, highligthed the Grand Canal Basin localization to localize the reader;
Response 4:We appreciate your valuable input, and we will improve the resolution of Figures 1, 2, and 3 to enhance clarity and better facilitate readers' understanding. Your feedback is greatly appreciated, and we are committed to delivering an improved manuscript.
Reviewer 2 Report
The manuscript exhibits a very good narrative sequence with a conveniently planned pace. English employed is remarkable.
The title refers to “spatio-temporal distribution characteristics”. This is quite controversial since spatial distribution is thoroughly addressed but no temporal considerations are made apart from the fact that data are contemporary. This should be seriously reconsidered since it creates a finally unsatisfied expectation in readers.
Literature employed is mostly Chinese and scarce international authors dealing with similar topics in other contexts such as Europe or America are not referenced. Some suggestions are provided later in order to better contextualize the manuscript.
Specifically in the different chapters:
1. Introduction:
The definition provided for Intangible Cultural Heritage should be as much general as possible. The first sentence provided “[…] represents the special crystallization on national civilization.” is too specific and has some political connotations which should be avoided. On the contrary, the definition after this first statement is a very good and inclusive one.
2. Research Data and Methods
Figure 1. The letter code for identifying the different areas is not the most appropriate and visual one. A color code being used would be appreciated since its understanding would be much easier.
3. Results and Analysis
Figure 2. A key should be provided with the meanings of R, Z and P.
Figure 3. The ten maps should be rearranged in such a way that a bigger scale can be provides. In its current state is hard to read the keys being a meaningful pity since it is perhaps the most relevant information and contribution of the paper.
Figure 4. Same comments as those provided for Figure 3.
3.5.1. Natural Geographic Factors: The statement “[…] In summary, the natural geographical environment has a certain influence on the formation of the spatial and temporal distribution characteristics of ICH in the Beijing-Hangzhou Grand Canal Basin, but the influence is not obvious.” Should be further developed. It is not just the influence of the natural geographical environment. If the territory analyzed exactly matches the basin is quite predictable that the spatial distribution of the ICH might match geometrically this area as well. The finding just proves that the ICH distribution is quite even and that there are no areas with a meaningful scarcity of ICH. In this sense, chapter 3.5.2 is much more sincere and objective.
4. Discussion and Conclusions
Statements such as “[…] On the whole, it has undergone an evolution process of “first concentration, then dispersion and then stability”. […]” are barely supported by the previous results and analysis. A better explanation is required.
References:
They are mostly quite local. It will be interesting to incorporate some more international references of recently published authors dealing with similar issues in other countries. Some suggestions:
Line 47, when dealing with the relationship between tourism and local and historical values:
Cabrera-i-Fausto, I., Fenollosa-Forner, E., & Serrano-Lanzarote, B. (2020). The new entrance to the Camí d’Onda Air-raid Shelter in the historic center of Borriana, Spain. TECHNE - Journal of Technology for Architecture and Environment, (19), 290-297. https://doi.org/10.13128/techne-7790
Santa-María-de-Andrés, I.C., & Cabrera-i-Fausto, I. (2021). El frontón Beti Jai de Madrid y su estructura. Tecnología, Diseño e Innovación, (7), 18-30. https://www.unae.edu.py/ojs/index.php/facat/article/view/333
Line 430, when dealing with the influence between factors such as natural geography, cultural ecology, socio-economic and political environment:
Casadei, C. (2022) The Inner Areas Italian question. The territory of Southern Inner Etruria along the via Clodia, ANUARI d’Arquitectura i Societat research journal (2), 134-166. ISSN: 2792-7601. https://doi.org/10.4995/anuari.2022.17945
Bartorila, M.A., & Loredo-Cansino, R. (2021) Cultural heritage and natural component. From reassessment to regeneration, ANUARI d’Arquitectura i Societat research journal, (1), 286-311. ISSN: 2792-7601. https://doi.org/10.4995/anuari.2021.16155
Author Response
Dear Reviewer,
Thank you for your valuable feedback on my manuscript. I have carefully read your suggestions and made corresponding revisions to the manuscript (Please see the attachment). Here are my responses to your suggestions:
Point 1:The title refers to “spatio-temporal distribution characteristics”. This is quite controversial since spatial distribution is thoroughly addressed but no temporal considerations are made apart from the fact that data are contemporary. This should be seriously reconsidered since it creates a finally unsatisfied expectation in readers.
Response 1:Thank you for raising the question. Our study focuses on the spatiotemporal distribution characteristics of intangible cultural heritage in the Grand Canal Basin. Regarding spatial features, we examine both the overall distribution and the classification of intangible cultural heritage. As for temporal analysis, we explore the timeline starting from the recognition of intangible cultural heritage. Our approach involves incorporating kernel density analysis, standard deviation ellipses, and mean center tools. The temporal nodes selected for analysis are 2006, 2008, 2011, 2014, and 2021. We sincerely appreciate your valuable feedback and suggestions, and we are committed to incorporating them into our manuscript.
Point 2:Literature employed is mostly Chinese and scarce international authors dealing with similar topics in other contexts such as Europe or America are not referenced. Some suggestions are provided later in order to better contextualize the manuscript.
Response 2:Thank you for your suggestions. We indeed have limited references to international literature. We will make the necessary modifications and additions based on your recommendations and the provided bibliography. Your feedback is highly appreciated, and we are committed to improving the comprehensiveness of our manuscript.
- Introduction:
The definition provided for Intangible Cultural Heritage should be as much general as possible. The first sentence provided “[…] represents the special crystallization on national civilization.” is too specific and has some political connotations which should be avoided. On the contrary, the definition after this first statement is a very good and inclusive one.
Response:Thank you for your suggestions. We will remove definitions with political implications and retain statements that effectively convey the definition of intangible cultural heritage. Your feedback is greatly appreciated, and we are committed to refining our manuscript accordingly.
- Research Data and Methods
Figure 1. The letter code for identifying the different areas is not the most appropriate and visual one. A color code being used would be appreciated since its understanding would be much easier.
Response:Thank you for your suggestion. We will take your advice into consideration and differentiate the four provinces and two cities in the basin using different colors.
- Results and Analysis
Figure 2. A key should be provided with the meanings of R, Z and P.
Response:I apologize for the confusion. I would like to kindly clarify that Figure 2 does not feature the elements R, Z, or P as mentioned.
Figure 3. The ten maps should be rearranged in such a way that a bigger scale can be provides. In its current state is hard to read the keys being a meaningful pity since it is perhaps the most relevant information and contribution of the paper.
Response:We appreciate your suggestions, and we will take them into consideration. Specifically, we will modify the layout of the image from 2 rows to 4 rows and enlarge it to enhance the clarity in presenting the spatial distribution characteristics of the top ten intangible cultural heritage categories. Thank you for your valuable input, and we are committed to improving the quality of the manuscript based on your feedback.
Figure 4. Same comments as those provided for Figure 3.
Response:Thank you very much for your feedback. Allow me to provide further clarification. Figure 3 showcases the kernel density analysis of diverse categories of intangible cultural heritage, while Figure 4 focuses on the kernel density analysis of intangible cultural heritage across different time periods.
3.5.1. Natural Geographic Factors: The statement “[…] In summary, the natural geographical environment has a certain influence on the formation of the spatial and temporal distribution characteristics of ICH in the Beijing-Hangzhou Grand Canal Basin, but the influence is not obvious.” Should be further developed. It is not just the influence of the natural geographical environment. If the territory analyzed exactly matches the basin is quite predictable that the spatial distribution of the ICH might match geometrically this area as well. The finding just proves that the ICH distribution is quite even and that there are no areas with a meaningful scarcity of ICH. In this sense, chapter 3.5.2 is much more sincere and objective.
Response:We wholeheartedly agree with your viewpoint. It is indeed worth addressing. In future research, we will conduct a more detailed analysis of the influence of topographic and geomorphic factors on the formation and development of intangible cultural heritage, taking into account the specific characteristics of the watershed. Our aim is to achieve a higher level of differentiation.
- Discussion and Conclusions
Statements such as “[…] On the whole, it has undergone an evolution process of “first concentration, then dispersion and then stability”. […]” are barely supported by the previous results and analysis. A better explanation is required.
Response:Thank you for your suggestion. This portion is obtained from temporal features, utilizing three spatial analysis tools: kernel density analysis, standard deviation ellipse, and mean center. These temporal features are reflected in the results analysis. We are considering incorporating additional charts and graphs to enrich the content analysis and validate the conclusions.
References:
They are mostly quite local. It will be interesting to incorporate some more international references of recently published authors dealing with similar issues in other countries. Some suggestions:
Response:Thank you very much for providing the literature. We will incorporate it into the manuscript and include it in the reference list.
Round 2
Reviewer 2 Report
Despite not fully considered, suggestions and ammendments provided by the reviewer have been sufficiently implemented.